# Housefly (*Musca domestica*) Larvae Preparations after Removing the Hydrophobic Fraction Are Effective Alternatives to Fish Meal in Aquaculture Feed for Red Seabream (*Pagrus major*)

**Atsushi Hashizume** [1], **Atsushi Ido** [1], **Takashi Ohta** [2], **Serigne Thierno Thiaw** [1], **Ryusaku Morita** [1], **Munenori Nishikawa** [1], **Takayuki Takahashi** [1], **Chiemi Miura** [1,3] **and Takeshi Miura** [1,*] 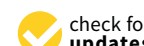

[1]   Graduate School of Agriculture, Ehime University, 3-5-7, Tarumi, Matsuyama, Ehime 790-8566, Japan
[2]   South Ehime Fisheries Research Center, Ehime University, 1289-1, Funakoshi, Ainan, Ehime 798-4292, Japan
[3]   Department of Global Environment Studies, Faculty of Environmental Studies, Hiroshima Institute of Technology, 2-1-1 Miyake, Saeki, Hiroshima 731-5193, Japan
\*   Correspondence: miutake@agr.ehime-u.ac.jp; Tel.: +81-89-946-3684

**Abstract:** Insects are an attractive alternative to fish meal (FM) as a sustainable protein source in aquaculture feed that does not negatively impact the marine ecosystem. Despite housefly (*Musca domestica*) larvae having adequacy of amino acid profiles, they have sometimes been reported to be inferior to FM, especially for marine carnivorous fish species. Here, we report that the removal of the hydrophobic fractions from housefly larvae enables significant replacement of FM in the diet of the red seabream (*Pagrus major*). In a feeding trial, housefly (HF) larvae that had the hydrophobic fraction removed as a complete substitution for 70% FM produced satisfactory growth. However, HF larvae that were supplemented with the hydrophobic fraction resulted in significant growth reduction. Growth recovery was incomplete by supplementation of docosahexaenoic acid (DHA) and eicosapentaenoic acid (EPA) to undefatted HF larvae, being equivalent to that of fatty acid content with a control diet. Moreover, fish with a dietary intake of catechol identified from the hydrophobic fraction of the HF showed growth reduction and morphological alterations in the intestine. Our findings indicate that the hydrophobic fraction from HF larvae contains a negative factor for fish growth and eliminating the fraction from HF larvae is thought to be an important process for sustainable aquaculture.

**Keywords:** sustainable aquaculture; fish meal replacement; insect for feed; catechol

## 1. Introduction

Certain types of aquaculture disrupt various marine ecosystems, in no small part due to the use of wild pelagic fish, e.g., sardines and anchovies, to produce fish meal (FM), particularly when farming marine carnivorous fishes [1]. Due to a rapid expansion of global aquaculture production, the development of alternative protein sources in feed for carnivorous fishes has become quite an urgent issue. Insects have attracted broad attention as a novel protein source for enhancing global food security since the Food and Agriculture Organization (FAO) published an assessment of edible insects as food and feed [2]. Similarly, the replacement of FM with insects in the feed for ornamental aquaculture was reported [3].

The housefly (*Musca domestica*) can yield animal protein in the biodegradation of organic waste [4], such as pig manure [5–10], cattle manure [11], poultry litter [12], and food waste [13]. The larvae

obtained from these systems are alternative feed ingredients that are rich in nutrients from within the human food chain without the need to harvest part of the wild marine ecosystem.

Although a number of studies have now reported on FM replacement in fish diets using housefly (HF) larvae, most of the studies were demonstrated with mainly freshwater-fish species, such as Nile tilapia (*Oreochromis niloticus*) and catfish species (*Clarias gariepinus* and *Heterobranchus longifil*) [14,15]. Another study demonstrated that a 30% inclusion of HF larvae meal in a diet was possible in barramundi (*Lates calcarifer*), but the authors concluded that up to 10% inclusion was recommended due to the physiological response in cultured fish caused by chitin, an imbalanced fatty acid profile (high $n-6$ fatty acids), or another unknown factor contained in HF larvae [16]. Substitution of FM with HF larvae seems to be limited, especially for marine carnivorous fish species, which have a high dietary protein requirement.

We previously reported that the low inclusion of HF (0.5–5%) in diets promoted growth and disease resistance for the marine carnivorous fish red seabream (*Pagrus major*) [17]; however, 25% or more inclusion of HF (50% or more replacement of FM) in diets resulted in growth reduction in red seabream (data not shown). While HF larvae appear to possess an adequate amino acid profile, similar to FM, the lack in high unsaturated omega$-3$ fatty acids was noted as a factor for the limit in their inclusion [18,19]. A negative factor in HF larvae for the growth of marine fish should be studied in detail for the establishment of sustainable aquaculture.

Insects are known to contain a variety of unique hydrophobic molecules [20]. One of these hydrophobic molecules that are unique to insects, catechol and its derivatives, which are known to be abundant in the cuticle, play a central role in the sclerotization and melanization by covalently inducing protein–protein and protein–chitin crosslinks [21,22]. An isomer and derivatives of catechol have been reported to exhibit acute toxicity in fish [23,24]. Although the dietary inclusion of the black soldier fly (*Hermetia illucens*) larvae was reported to affect enzymatic activities [25] and intestinal microbiota [26] in the laying hen, the dietary effect of these kinds of molecules from insects on cultured fish growth and their intestines remains unknown.

In the current study, we report that the removal of the hydrophobic fraction from HF larvae enables more significant replacement of FM in the diet of the red seabream. Moreover, the free catechol involved in insect cuticular was detected in the hydrophobic fraction from HF larvae. Although no significant difference in growth was observed with the low dietary level of catechol, morphology alterations were observed in the intestine of red seabream. This study indicates that catechol in HF larvae is thought to be one of the negative factors causing growth reduction in cultured marine fish.

## 2. Results

### 2.1. Amino Acid and Fatty Acid Compositions of HF Larvae

To assess whether HF larvae are a truly viable alternative protein source in aquaculture feeds, we first analyzed their amino acid profiles. HF larvae contained all of the amino acids found in FM, including the 10 essential amino acids, and also tyrosine and cysteine (Table 1), which are commonly required for the growth and survival of fish [15,27]. The amino acid balance of HF larvae was comparable to those of FM. However, the quantity of taurine was only 0.3 mg/g in HF larvae, compared to 1.5 mg/g in FM.

**Table 1.** Amino acid composition in fish meal and housefly larvae.

|  | **Fish Meal (FM) (CP65)** | **Housefly (HF) Larvae** |
|---|---|---|
| Essential amino acids (%) | | |
| Arg | 4.25 (6.52) | 2.91 (5.52) |
| Lys | 5.38 (8.26) | 4.21 (7.98) |
| His | 2.18 (3.35) | 1.63 (3.09) |
| Phe | 2.83 (4.34) | 3.81 (7.22) |
| Leu | 5.35 (8.21) | 3.41 (6.46) |
| Ile | 2.82 (4.33) | 2.08 (3.94) |
| Met | 2.01 (3.08) | 1.40 (2.65) |
| Val | 3.37 (5.17) | 2.80 (5.31) |
| Thr | 2.97 (4.56) | 2.28 (4.32) |
| Trp | 0.83 (1.27) | 0.71 (1.35) |
| Nonessential amino acids (%) | | |
| Tyr | 2.12 (3.25) | 4.14 (7.85) |
| Ala | 4.58 (7.03) | 2.78 (5.27) |
| Gly | 5.03 (7.72) | 2.33 (4.42) |
| Pro | 3.23 (4.96) | 2.27 (4.30) |
| Glu | 8.57 (13.15) | 7.71 (14.61) |
| Ser | 2.78 (4.27) | 2.18 (4.13) |
| Asp | 6.22 (9.54) | 5.57 (10.56) |
| Cys–Cys | 0.65 (1.00) | 0.53 (1.00) |
| Total (%) | 65.2 (100) | 52.75 (100) |
| Taurine (mg/g dry matter) | 1.5 | 0.3 |

Normalized percentiles of amino acids with total proteins are indicated in ( ). Amino acids of fish meal were based on FM with 65% crude protein in the Standard Table of Feed Composition in Japan (National Agriculture and Food Research Organization, 2009) [28].

Marked differences were found between the fatty acid composition of fish oil and HF larva (Table 2). The total n−3 fatty acid level was 0.4% in HF larvae in comparison with 27.3% in fish oil. In addition, the only n−3 fatty acids present in the larvae was linolenic acid (18:3n−3). Notably, housefly larvae also entirely lacked n−3 highly unsaturated fatty acids (n−3 HUFAs), which are designated as the n-3 fatty acids with 20 or more carbons and are considered essential nutrients for marine fish [29].

**Table 2.** Fatty acid composition in fish oil and housefly larva.

| **Components (% Fatty Acids)** | **Fish Oil** | **HF Larva** |
|---|---|---|
| Saturated fatty acid | | |
| 12:0 | 4.1 | 0.1 |
| 14:0 | | 3.1 |
| 15:0 | 0.5 | 2.3 |
| 16:0 | 15.6 | 24.8 |
| 17:0 | 0.7 | 0.9 |
| 18:0 | 3.6 | 5.0 |
| 20:0 | 0.3 | 0.2 |
| 22:0 | | 0.1 |
| Total | 24.8 | 36.5 |

**Table 2.** *Cont.*

| Components (% Fatty Acids) | Fish Oil | HF Larva |
|---|---|---|
| Monounsaturated fatty acid | | |
| 14:1 | | 0.3 |
| 16:1 | 5.0 | 23.9 |
| 17:1 | 0.5 | 0.5 |
| 18:1 | 19.4 | 19.5 |
| 20:1 | 5.2 | |
| 22:1 | 5.0 | |
| 24:1 | 0.6 | |
| Total | 35.7 | 44.2 |
| Polyunsaturated fatty acid | | |
| n−3 fatty acid | | |
| 18:3n−3 | 1.0 | 0.4 |
| 20:3n−3 | 0.2 | |
| 20:4n−3 | 0.7 | |
| 20:5n−3 | 7.5 | |
| 21:5n−3 | 0.3 | |
| 22:5n−3 | 1.9 | |
| 22:6n−3 | 15.7 | |
| Total | 27.3 | 0.4 |
| n−6 fatty acid | | |
| 18:2n−6 | 2.8 | 6.1 |
| 20:2n−6 | 0.3 | |
| 20:3n−6 | 0.2 | |
| 20:4n−6 | 1.1 | |
| 22:5n−6 | 0.6 | |
| Total | 5.0 | 6.1 |
| Others | | |
| 16:2 | 0.3 | |
| 16:3 | 0.2 | 0.2 |
| 16:4 | 0.3 | 0.2 |
| Total | 0.8 | 0.4 |
| Not identified | 4.3 | 12.4 |

*2.2. Growth Performance in Red Seabream with Diets That Completely Replaced FM with HF Larvae*

To evaluate the dietary effects of using HF larvae in fish diets, with or without their hydrophobic fraction, feeding trials were conducted for red seabream. Dried pellets containing 70% FM or 70% HF larvae (complete replacement of FM), with or without the hydrophobic fraction (HF$^{+HPF}$ and HF$^{-HPF}$), were fed to the fish. The results indicated no significant differences in fork-length (FL) and body-weight (BW), FL and BW gain, and FL and BW gain percentages between the 70% HF$^{-HPF}$ and 70% FM groups (Figure 1; Table 3). However, the 70% HM$^{+HPF}$ group showed significantly reduced BW at two weeks and four weeks in comparison with 70% FM group, and FL and BW gain percentages in comparison with the other two groups. When we measured the feed intake and feed conversion ratio, the 70% HF$^{-HPF}$ group again did not show clear differences from the 70% FM diet group, but the 70% HF$^{+HPF}$ group showed poorer outcomes (Table 3). We conducted another feeding trial, comparing defatted HF with undefatted HF diets. The experimental diets in which 40% FM in the control diet was again completely substituted with defatted HF larvae showed similar growth performance results in FL and BW gain, and FL and BW gain percentages (Table 4) in the feeding trial 1. The diet with 40% undefatted HF resulted in reduced BW and FL gain. In the feeding trial 2, although 40% HF$^{-HPF}$ and 40% undeffatted HF showed similar results as the feeding trial 1, notably, growth recovery, especially in terms of BW gain, was incomplete by supplementation of the 40% undefatted

HF diet with docosahexaenoic acid (DHA) and eicosapentaenoic acid (EPA) (Table 5). Almost all of the fish survived to the end of the trials (Tables 3–6). All deaths that occurred were determined to have been accidental.

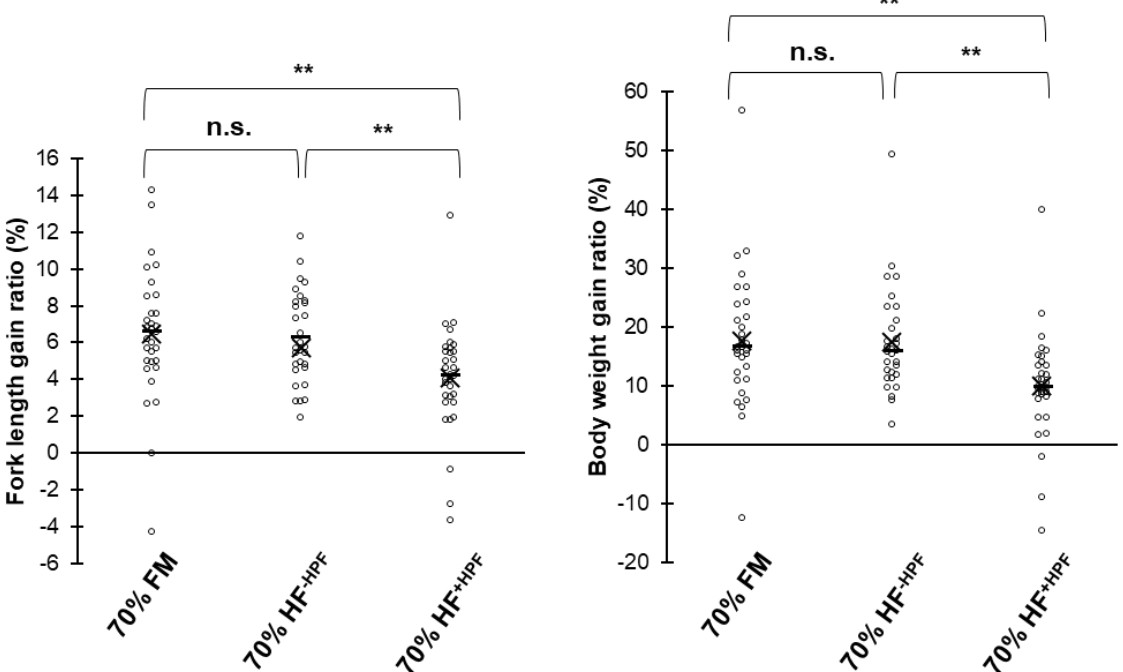

**Figure 1.** Growth performance of red seabream on feeds containing 70% Housefly larvae with or without the hydrophobic fraction. **Left**, fork length gain rate; **right**, body weight gain rate. Fish were fed with 70% fish meal diet (70% FM), 70% housefly larvae removed hydrophobic fraction diet (70% HF$^{-HPF}$) or 70% housefly larvae supplemented with the hydrophobic fraction diet (70% HF$^{+HPF}$); ×, mean value; horizontal bar, median value; The 70% FM, 70% HF$^{-HPF}$, and HF$^{+HPF}$ dietary groups were statistically analyzed using the Steel–Dwass multiple comparison test as a post hoc test after the Kruskal–Wallis test ($p < 0.05$); n.s.: not significant; **$p < 0.01$.

**Table 3.** Growth performance of red seabream with complete replacement of 70% FM with HF larvae.

| Parameters | Week | Diet Group | | |
| --- | --- | --- | --- | --- |
| | | 70% FM | 70% HF$^{-HPF}$ | 70% HF$^{+HPF}$ |
| Growth data | | | | |
| *n* | | 31 | 30 | 32 |
| FL (cm) | Initial | 10.5 (10.3, 10.6) [a] | 10.4 (10.1, 10.6) [a] | 10.4 (10.2, 10.6) [a] |
| | 2 weeks | 10.8 (10.7, 11.0) [a] | 10.7 (10.5, 11.0) [a] | 10.7 (10.5, 10.9) [a] |
| | 4 weeks | 11.1 (11.0, 11.3) [a] | 11.0 (10.7, 11.3) [a] | 10.9 (10.6, 11.1) [a] |
| BW (g) | Initial | 25.2 (24.0, 26.3) [a] | 23.6 (22.1, 25.0) [a] | 23.5 (22.2, 24.8) [a] |
| | 2 weeks | 27.3 (26.0, 28.6) [a] | 25.2 (23.6, 26.8) [a,b] | 24.2 (22.8, 25.6) [b] |
| | 4 weeks | 29.5 (28.0, 31.0) [a] | 27.6 (25.8, 29.4) [a,b] | 25.8 (24.2, 27.3) [b] |
| FL gain (cm) | Initial–4 weeks | 0.7 (0.5, 0.8) [a] | 0.6 (0.5, 0.7) [a] | 0.4 (0.3, 0.5) [b] |
| BW gain (g) | Initial–4 weeks | 4.3 (3.2, 5.5) [a] | 4.1 (3.3, 4.9) [a] | 2.3 (1.5, 3.0) [b] |
| FL gain percentage (%) | Initial–4 weeks | 6.4 (5.1, 7.8) [a] | 6.2 (5.3, 7.2) [a] | 4.0 (3.0, 5.1) [b] |
| BW gain percentage (%) | Initial–4 weeks | 17.7 (13.4, 21.9) [a] | 17.4 (14.1, 20.7) [a] | 9.9 (6.6, 13.2) [b] |
| Total feed intake per fish (g) | 4 weeks | 8.5 | 7.9 | 6.0 |
| Feed conversion ratio | 4 weeks | 2.03 | 2.07 | 2.65 |
| Survival rate (%, n/N) | 4 weeks | 96.9 (31/32) | 93.8 (30/32) | 100 (32/32) |

Abbreviations: FL, fork length; BW, body weight; 70% FM, 70% fish meal diet; 70% HF$^{-HPF}$, 70% housefly larvae removed hydrophobic fraction diet; 70% HF$^{+HPF}$, 70% housefly larvae supplemented with the hydrophobic fraction diet. Data are represented as means (upper limit and lower limit of the 95% confidential interval). Different letters indicate statistically significant differences according to the Steel–Dwass multiple comparison test as a post hoc test after the Kruskal–Wallis test ($p < 0.05$).

**Table 4.** Growth performance of red seabream with complete replacement of 40% FM with HF larvae in feeding trial 1.

| | | Diet Group | | |
|---|---|---|---|---|
| **Parameters** | **Week** | **40% FM** | **40% HF$^{-HPF}$** | **40% Undefatted HF** |
| Growth data | | | | |
| n | | 39 | 40 | 39 |
| FL (cm) | Initial | 11.5 (11.4, 11.6) [a] | 11.7 (11.5, 11.8) [a] | 11.6 (11.5, 11.8) [a] |
| | 2 weeks | 11.6 (11.4, 11.7) [a] | 11.7 (11.6, 11.9) [a] | 11.6 (11.5, 11.8) [a] |
| | 4 weeks | 11.6 (11.4, 11.8) [a,b] | 11.7 (11.6, 11.9) [b] | 11.5 (11.3, 11.6) [a] |
| BW (g) | Initial | 29.0 (28.2, 29.9) [a] | 30.0 (29.2, 30.9) [a] | 29.3 (28.4, 30.1) [a] |
| | 2 weeks | 29.1 (27.6, 30.6) [a] | 30.9 (29.7, 32.1) [a] | 30.1 (29.0, 31.3) [a] |
| | 4 weeks | 32.5 (29.9, 35.2) [a] | 33.5 (31.8, 35.2) [a] | 31.4 (30.2, 32.6) [a] |
| FL gain (cm) | Initial–4 weeks | 0.1 (−0.1, 0.3) [a,b] | 0.1 (−0.1, 0.2) [a] | −0.2 (−0.3, 0.0) [b] |
| BW gain (g) | Initial–4 weeks | 3.5 (1.3, 5.7) [a] | 3.4 (2.2, 4.7) [a] | 2.1 (1.5, 2.8) [a] |
| FL gain percentage (%) | Initial–4 weeks | 1.0 (−0.6, 2.6) [a,b] | 0.7 (−0.8, 2.2) [a] | −1.3 (−2.2, −0.4) [b] |
| BW gain percentage (%) | Initial–4 weeks | 11.5 (4.2, 18.8) [a] | 11.2 (7.2, 15.2) [a] | 7.2 (5.0, 9.3) [a] |
| Total feed intake per fish (g) | 4 weeks | 7.1 | 7.4 | 6.1 |
| Feed conversion ratio | 4 weeks | 2.01 | 2.15 | 2.88 |
| Survival rate (%, n/N) | 4 weeks | 97.5 (39/40) | 100 (40/40) | 97.5 (39/40) |

Abbreviations: FL, fork length; BW, body weight; 40% FM, 40% fish meal diet; 40% HF$^{-HPF}$, 40% housefly larvae removed hydrophobic fraction diet; 40% Undefatted HF. 40% undefatted housefly larvae diet. Data are represented as means (upper limit and lower limit of the 95% confidential interval) in FL, BW, FL gain, BW gain, FL gain rate, and BW gain rate. Different letters indicate statistically significant differences according to the Steel–Dwass multiple comparison test as a post hoc test after the Kruskal–Wallis's test ($p < 0.05$).

**Table 5.** Growth performance of red seabream with complete replacement of 40% FM with HF larvae in feeding trial 2.

| | | Diet Group | | | |
|---|---|---|---|---|---|
| **Parameters** | **Week** | **40% FM** | **40% HF$^{-HPF}$** | **40% Undefatted HF** | **40% Undefatted HF$^{+DHA+EPA}$** |
| Growth data | | | | | |
| n | | 21 | 21 | 22 | 22 |
| FL (cm) | Initial | 12.7 (12.2, 13.3) [a] | 12.8 (12.4, 13.2) [a] | 12.9 (12.4, 13.4) [a] | 12.8 (12.3, 13.2) [a] |
| | 2 weeks | 13.3 (12.8, 13.8) [a] | 13.3 (12.9, 13.7) [a] | 13.3 (12.7, 13.8) [a] | 13.3 (12.8, 13.7) [a] |
| | 4 weeks | 14.2 (13.7, 14.8) [a] | 14.3 (13.9, 14.6) [a] | 13.8 (13.3, 14.4) [a] | 14.0 (13.4, 14.6) [a] |
| BW (g) | Initial | 47.2 (41.7, 52.7) [a] | 48.2 (42.3, 54.0) [a] | 50.3 (44.7, 55.8) [a] | 48.5 (43.2, 53.9) [a] |
| | 2 weeks | 53.2 (47.1, 59.3) [a] | 52.8 (46.1, 59.4) [a] | 52.1 (46.0, 58.2) [a] | 53.2 (47.2, 59.3) [a] |
| | 4 weeks | 69.7 (61.2, 78.3) [a] | 66.3 (58.3, 74.4) [a] | 60.2 (52.6, 67.9) [a] | 61.8 (54.6, 69.0) [a] |
| FL gain (cm) | Initial–4 weeks | 1.5 (1.3, 1.7) [a] | 1.5 (1.3, 1.7) [a] | 0.9 (0.7, 1.1) [b] | 1.2 (1.0, 1.5) [a,b] |
| BW gain (g) | Initial–4 weeks | 21.9 (17.0, 26.9) [a] | 18.2 (14.2, 22.1) [a,b] | 11.1 (7.3, 14.9) [b] | 13.2 (9.2, 17.2) [a,b] |
| FL gain percentage (%) | Initial–4 weeks | 11.6 (9.3, 14.0) [a] | 11.9 (9.6, 14.2) [a] | 7.5 (5.8, 9.1) [b] | 9.6 (7.6, 11.6) [a,b] |
| BW gain percentage (%) | Initial–4 weeks | 48.0 (37.5, 58.5) [a] | 40.7 (30.1, 51.3) [a] | 22.9 (14.8, 31.1) [b] | 27.9 (19.6, 36.2) [a,b] |
| Total feed intake per fish (g) | 4 weeks | 30.8 | 30.4 | 23.6 | 26.6 |
| Feed conversion ratio | 4 weeks | 1.41 | 1.67 | 2.13 | 2.01 |
| Survival rate (%, n/N) | 4 weeks | 95.4 (21/22) | 95.4 (21/22) | 100 (22/22) | 100 (22/22) |

Abbreviations: FL, fork length; BW, body weight; DHA, docosahexaenoic acid; EPA, eicosapentaenoic acid. Data are represented as means (upper limit and lower limit of the 95% confidential interval) in FL, BW, FL gain, BW gain, FL gain rate, and BW gain rate. Different letters indicate statistically significant differences according to the Steel–Dwass multiple comparison test as a post hoc test after the Kruskal–Wallis test ($p < 0.05$).

**Table 6.** Growth performance of red seabream with diets with added catechol.

| | | Diet Group | | | |
|---|---|---|---|---|---|
| **Parameters** | **Week** | **Control** | **20 µg/g** | **200 µg/g** | **2000 µg/g** |
| Growth data | | | | | |
| n | | 29 | 27 | 24 | 23 |
| FL (cm) | Initial | 5.7 (5.6, 5.8) [a] | 5.6 (5.4, 5.8) [a] | 5.7 (5.5, 5.9) [a] | 5.6 (5.4, 5.8) [a] |
| | 2 weeks | 6.7 (6.5, 6.9) [a] | 6.7 (6.5, 6.9) [a] | 6.8 (6.5, 7.0) [a] | 5.6 (5.4, 5.8) [b] |
| | 4 weeks | 8.5 (8.3, 8.7) [a] | 8.2 (8.0, 8.4) [a] | 8.2 (7.9, 8.4) [a] | 5.7 (5.5, 5.9) [b] |
| BW (g) | Initial | 3.2 (2.9, 3.5) [a] | 3.3 (3.0, 3.6) [a] | 3.3 (3.0, 3.6) [a] | 3.2 (2.9, 3.5) [a] |
| | 2 weeks | 6.7 (6.5, 6.9) [a] | 6.4 (5.8, 7.0) [a] | 6.5 (5.8, 7.2) [a] | 3.1 (2.8, 3.4) [b] |
| | 4 weeks | 13.7 (12.6, 14.8) [a] | 12.6 (11.4, 13.8) [a] | 12.4 (11.1, 13.7) [a] | 3.3 (2.9, 3.7) [b] |
| FL gain (cm) | Initial–4 weeks | 2.8 (2.6, 3.0) [a] | 2.5 (2.3, 2.7) [a] | 2.5 (2.2, 2.8) [a] | 0.1 (−0.1, 0.3) [b] |
| BW gain (g) | Initial–4 weeks | 10.5 (9.4, 11.6) [a] | 9.3 (8.1, 10.5) [a] | 9.1 (7.8, 10.4) [a] | 0.1 (−0.3, 0.5) [b] |
| FL gain percentage (%) | Initial–4 weeks | 49.3 (45.8, 52.9) [a] | 44.7 (40.3, 49.1) [a] | 43.5 (38.9, 48.1) [a] | 2.6 (−1.2, 6.4) [b] |
| BW gain percentage (%) | Initial–4 weeks | 323.3 (289.1, 357.5) [a] | 284.6 (247.1, 322.1) [a] | 277.4 (239.0, 315.8) [a] | 1.9 (−10.3, 14.1) [b] |
| Total feed intake per fish (g) | 4 weeks | 8.1 | 7.2 | 7.3 | 1.1 |
| Feed conversion ratio | 4 weeks | 0.77 | 0.77 | 0.77 | 11.0 |
| Survival rate (%, n/N) | 4 weeks | 96.7 (29/30) | 90.0 (27/30) | 80.0 (24/30) | 76.7 (23/30) |

Abbreviations: FL, fork length; BW, body weight. Data are represented as means (upper limit and lower limit of the 95% confidential interval) in FL, BW, FL gain, BW gain, FL gain rate, and BW gain rate. Different letters indicate statistically significant differences according to the Steel–Dwass multiple comparison test as a post hoc test after the Kruskal–Wallis test ($p < 0.05$).

*2.3. Detection of Catechol from HF Larvae and Evaluation of Dietary Catechol in Red Seabream*

We confirmed the free catechol concentration in HF larvae by Gas Chromatography-Mass spectrometry (GC/MS) before using insect matter in subsequent feeding trials. Although the concentration of free catechol was 27.1 µg/g in HF larvae before any treatment, that was undetected (<1 µg/g) in defatted HF larvae after treatment with hexane + ethanol (9:1). Moreover, we conducted a feeding trial to evaluate the effect of catechol dietary intake on juvenile red seabream. As Table 6 shows, the survival rate in each group declined according to catechol concentration. Although there was no significant difference at four weeks of feeding, fish fed with an addition of 20 and 200 µg/g catechol resulted in a slight reduction in FL and BW compared to the fish in the control (no addition of catechol). Notably, red seabream fed with 2000 µg/g catechol showed almost no growth in the trial. In histological observations of the intestine sections, morphological alterations in the intestines of red seabream occurred in accordance with catechol concentration (Figure 2). In particular, microvillus and epithelial-cell heights, and areas of goblet cells in fish with a dietary intake of catechol were significantly reduced compared to the controls (Figure 3).

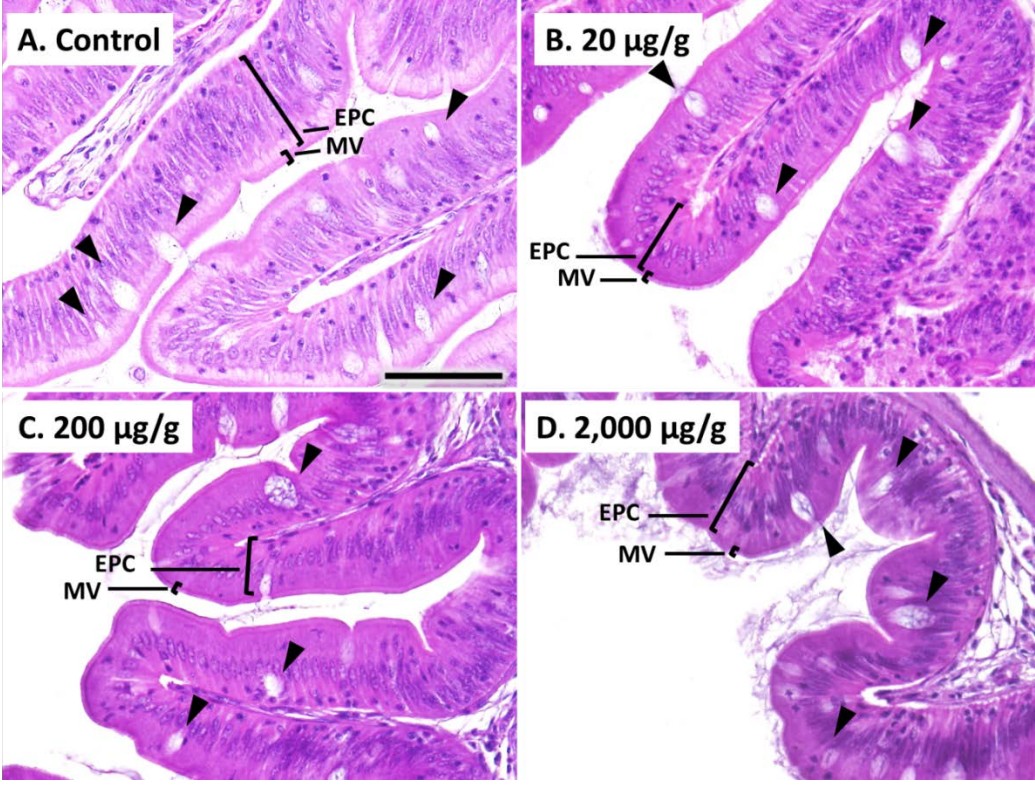

**Figure 2.** Histological sections of intestines from red seabream in (**A**) control group; (**B**) 20 µg/kg catechol group; (**C**) 200 µg/g catechol group; and (**D**) 2000 µg/g catechol group; EPC, epithelial cell; MV, microvillus; Arrowheads indicate goblet cells; Scale bar, 50 µm.

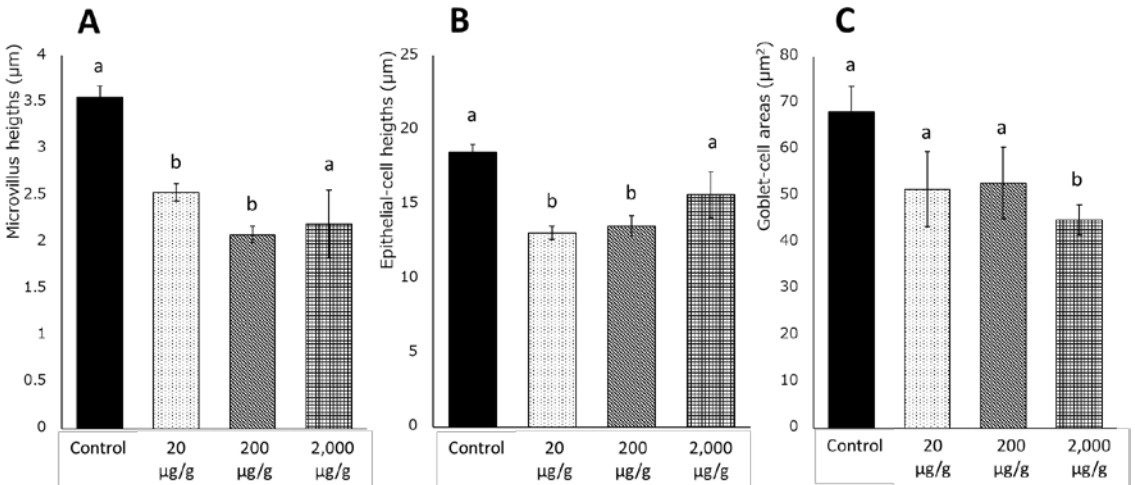

**Figure 3.** Effects of dietary intake of catechol on (**A**) microvillus heights; (**B**) epithelial-cell heights; and (**C**) goblet-cell areas in red seabream intestine. Data are expressed as means, and error bars represent standard errors. Different letters (a, b) indicate statistically significant differences according to the Steel–Dwass multiple comparison tests as a post hoc test after the Kruskal–Wallis test ($p < 0.05$).

## 3. Discussion

We found in our present study that FM can be successfully replaced with HF larvae in an aquaculture feed far beyond the 25% level if the hydrophobic fraction is removed with an organic-solvent elimination process. A large amount of FM in diets could be replaced with HF larvae preparation after the removal of hydrophobic fractions, and our findings show the potential of HF larvae as an alternative feed ingredient for sustainable aquaculture production. Although the cost of the hydrophobic fraction removal must be considered, the purchasing price of dried HF larvae produced from swine manure in a full-scale commercial system in China was reported to be $1.25 to US$1.43 kg$^{-1}$ [8]; the annual average of the FM price in 2018 was US$1.49 kg$^{-1}$ [30]; therefore, HF larvae could economically substitute FM. The amino acid profile in HF larvae was comparable with that of FM except for taurine. Lack of dietary taurine resulted in green-liver syndrome and growth retardation [31–33]. Hence, we determined that HF larvae are a potentially viable protein source for farmed fish in their amino acids, but that supplementation with taurine would be required for their use in aquaculture feeds.

With respect to fatty acids, n−3 high unsaturated fatty acids (HUFAs) are absent in HF larvae but could be added as a supplement to the diet. However, recovery of fish growth was partial, and improvements in BW gain and feed intake were insufficient. A substantial portion (12.4%) of total fatty acids were not identified in this study. Unidentified fatty acids in HF larvae or pupae were relatively high (7.2–8.5%) comparing with other insects [19]. Since the composition of unidentified fatty acids in HF larvae reached 15% and depended on sources of poultry litter as rearing substrates for HF larvae [12], these unidentified fatty acids seemed to be derived from pig manure used for rearing the larvae. Chitin, a linear homopolymer of β(1-4)-linked *N*-acetylglucosamine units, a major constituent of insect cuticles, was thought to influence growth reduction in marine fish because of its difficulty in digestion [34]. Although 3.0–6.8% nitrogen of the total nitrogen was thought to be chitin in insect larvae [35], chitin in diets could be digested and utilized in red seabream, as the growth of red seabream was improved when diets were supplemented with 10% chitin [36]. Chitinase activity in fish stomach is known to degrade chitin efficiently [37]. Therefore, HF larvae in diets may contain other factors to prevent growth in the red seabream because chitin from HF larvae could be utilized by the red seabream.

Insects are known to contain a variety of unique hydrophobic molecules [20]. These molecules in the hydrophobic fractions of HF larvae, other than the nutritional deficiency caused by the absence of n-3 HUFAs, could also participate in growth reduction. Since the dietary intake of a hydrophobic

fraction from yellow-mealworm (*Tenebrio molitor*) larvae resulted in significant growth reduction in red seabream [38], this property could be observed not only in HF larvae but in a wide range of insect species. One explanation for the reduced growth is the involvement of insect cuticular molecules, such as straight or methyl-branched n-alkanes, n-alkenes, fatty acids, alcohols, wax esters [39,40], and catechol and its derivatives [22,23] in the hydrophobic fractions. In particular, hydroquinone (1, 3-benzendiol), a structural isomer of catechol, was reported to exhibit acute toxicity in fathead minnow (*Pimephales promelas*) and rainbow trout (*Oncorhynchus mykiss*) [23]; toxicity of catechol derivatives, such as dichlorocatechol, was also known in brown trout (*Salmo trutta*) [24] in exposure assessments. The mechanisms of their toxicity were thought to be the generation of reactive oxygen species (ROS), DNA and protein damage, or interference with electron transport in energy-transducing membranes [41]. In the current study, red seabreams with a dietary intake of catechol showed intestinal morphological alterations at lower doses and growth reductions at higher doses. These findings suggest that catechol in the hydrophobic fraction of HF larvae causes incomplete growth recovery despite supplementation with DHA and EPA in diets. In HF larvae, 27.1 μg/g catechol was detected before the removal of hydrophobic fractions, and approximately 20 μg/g catechol was estimated to be contained in the 70% HF$^{+HPF}$ diet. Since no significant difference in growth after four weeks of feeding was observed with the addition of 20 μg/g and 200 μg/g catechol, dietary intake of other hydrophobic molecules than catechol in the fractions of HF larvae seems to affect fish growth. Nevertheless, regression of microvillus, intestinal epithelium cells, and goblet cells, which have important roles in nutrition absorption, were observed in the intestine of red seabream with the dietary intake of 20 μg/g catechol; thus, these results imply that strain on the intestine caused by dietary catechol even at low levels resulted in partial growth reduction. As we showed in the previous study, the diet with a low inclusion of HF gave disease resistance in red seabream [17]. In contrast, since intestinal epithelial cells in fish are related to innate immunity, and the recognition of pathogens [42], morphological alterations in the intestine by the dietary intake of catechol might provoke infectious diseases. Although further observation of the intestine with another method, such as Fourier Transform Infrared (FTIR) spectroscopy [43] is needed, removal of the hydrophobic fraction including catechol was thought to be an essential process for the high inclusion of HF larvae in diets.

Importantly, without defatting, it is not easy to use HF larvae in a marine-fish diet and obtain appropriate crude protein and lipid composition, since HF larvae have an excess of crude lipids (20–35%, dry matter) and markedly different fatty acid profiles than feeds made from wild fish. The fatty acid profile in black-soldier fly larvae (*Hermetia illucens*) was reported to affect lipid composition and accumulation in zebrafish (*Danio rerio*) [44,45]. Eliminating the hydrophobic faction is, therefore, an important first step in the development of HF larva-based feeds for marine-fish aquaculture.

Long-term feeding trials of different HF larva formulations are still needed to further evaluate the growth performance and safety of the hydrophobic fraction of HF larvae in farmed fish. Other bioactive molecules than catechol in the hydrophobic fraction should be studied, and the upper limits of the HF larvae of fish feed after the removal of the hydrophobic fraction should also be elucidated in other marine-fish species. Furthermore, the development of defatting methods that do not require organic solvents is desirable for practical reasons. However, our present study provides some key new findings; the replacement of FM with HF larvae in diets for marine carnivorous fish like red seabream is a potentially viable option that could lead to a more environmentally sustainable aquaculture industry in the future.

## 4. Materials and Methods

### 4.1. Feed Ingredients

HF larvae were reared with pig manure by E's Inc. (Tokyo, Japan), boiled for 15 min and frozen at –30 °C until use. HF larvae were ground in a mill mixer after air-drying at 80 °C for 5 h or more. To defat the HF larvae, the ground HF larvae were treated twice with a four-fold volume of normal hexane

+ ethanol (9:1) for 2 h at room temperature. This defatting process was repeated several times until the crude lipids reached below 10% (Table 7). The supernatant in the defatting process was removed after centrifugation, and the sediment was air-dried at 60 °C for one day or more until completely air-dried. The HF hydrophobic fractions (HPF) were prepared from the aforementioned supernatant containing hexane + ethanol (9:1) by evaporation at room temperature until the volume was decreased by about 10-fold; then the temperature was gradually increased from 40 to 80 °C until the residual organic solvents had evaporated. FM from Peruvian anchoveta (*Engraulis ringens*) comprising 65% crude protein and cod fish oil were obtained from the Shintoa Corporation (Tokyo, Japan). DHA and EPA were kindly provided by Bizen Chemical Co., Ltd. (Akaiwa, Okayama, Japan).

**Table 7.** Proximate composition of housefly larvae before or after the removal of the hydrophobic fraction.

| Proximate Composition (%) | Fish Meal | HF Larvae | Defatted HF Larvae |
|---|---|---|---|
| Moisture | 8.97 | 7.24 | 3.91 |
| Crude protein | 64.9 | 48.30 | 67.0 |
| Crude lipid | 8.88 | 29.56 | 8.54 |
| Ash | 14.92 | 4.98 | 5.97 |
| NFE + crude fiber * | 2.33 | 9.92 | 14.58 |

* Nitrogen-free element (NFE) + crude fiber = 100 − (moisture + crude protein + crude lipid + ash).

## 4.2. Experimental Diets

We tried the complete replacement of 70% FM and 40% FM with HF larvae. In the first trial, a control diet included 70% FM, and FM in the diet was replaced with 70% defatted HF larvae (100% replacement of FM). Fish oil was added at 7% to the control diet and HF diet (HF$^{-\mathrm{HPF}}$), and the hydrophobic fraction from HF was added to another HF diet (HF$^{+\mathrm{HPF}}$) as an alternative to fish oil (Table 8). In the second study, the control diet included 40% FM and 11% fish oil, and the FM in the diet was replaced with 40% defatted HF larvae and fish oil (HF$^{-\mathrm{HPF}}$). We produced two more diets where 40% FM in the control diet was replaced with undefatted HF larvae, and we added DHA and EPA to the one diet to be equivalent to n-3 HUFA content levels among the control, HF$^{-\mathrm{HPF}}$, and undefatted HF$^{+\mathrm{DHA+EPA}}$ (Table 9). Crude protein in all these experimental diets was targeted to be around 50% to obtain the satisfactory growth of red seabream [46]. Supplementation with plant-derived protein sources (soybean meal and corn gluten meal) was done to achieve this protein level. To prepare dried pellets for use in the feeding trials, the powdered forms of all components were thoroughly mixed and supplemented with fish oil. These mixtures were then granulated after adding water and air-dried at 60 °C for more than one day. The resulting dry pellets were stored at 4 °C until use. In the feeding trial for the evaluation of catechol, catechol (Nakalai Tesque, Kyoto, Japan; ≥99.0% GC assay) was added to commercially available extruded pellet diets for juvenile red seabream Otohime$^{\mathrm{TM}}$ (Marubeni Nisshin Feed Co., Ltd., Tokyo, Japan). Catechol was added to the diets at 20, 200, and 2000 μg/g, and the surface of the diets was coated with plant-derived oil.

**Table 8.** Formulation and proximate composition of the 70% HF diets in the feeding trial.

| Ingredients (%) | Diet Group | | |
|---|---|---|---|
| | Control | HF$^{-HPF}$ | HF$^{+HPF}$ |
| FM | 70 | | |
| HF larvae (defatting) | | 70 | 70 |
| Fish oil | 7 | 7 | |
| HF hydrophobic fraction | | | 7 |
| Starch | 12 | 12 | 12 |
| Vitamin mix | 0.8 | 0.8 | 0.8 |
| Mineral mix | 0.4 | 0.4 | 0.4 |
| Taurine | 1.5 | 1.5 | 1.5 |
| Stabilized vitamin C | 0.05 | 0.05 | 0.05 |
| Choline chloride | 0.05 | 0.05 | 0.05 |
| Sodium dihydrogen phosphate | 0.8 | 0.8 | 0.8 |
| Potassium dihydrogen phosphate | 0.8 | 0.8 | 0.8 |
| Calcium lactate | 1.5 | 1.5 | 1.5 |
| Carboxymethyl cellulose | 5 | 5 | 5 |
| Proximate component (%) | | | |
| Moisture | 3.16 | 4.19 | 4.28 |
| Crude protein | 51.20 | 48.90 | 53.91 |
| Crude lipid | 13.78 | 12.84 | 14.56 |
| Ash | 13.93 | 7.79 | 7.21 |
| NFE + crude fiber * | 17.93 | 26.28 | 20.01 |

* NFE + crude fiber = 100 − (moisture + crude protein + crude lipid + ash).

**Table 9.** Formulation and proximate composition of the 40% HF diets in feeding trials.

| Ingredients (%) | Diet Group | | | |
|---|---|---|---|---|
| | Control | HF$^{-HPF}$ | Undefatted HF | Undefatted HF$^{+DHA+EPA}$ |
| FM | 40 | | | |
| HF larvae (defatting) | | 40 | | |
| HF larvae (no defatting) | | | 40 | 40 |
| Fish oil | 11 | 12 | | |
| DHA | | | | 0.9 |
| EPA | | | | 1.9 |
| Starch | 6 | 6 | 6 | 6 |
| Wheat flour | 10.2 | 4.6 | 7.1 | 4.3 |
| Soybean meal | 16 | 16 | 16 | 16 |
| Corn gluten meal | 13 | 17.5 | 27 | 27 |
| Vitamin mix | 0.4 | 0.4 | 0.4 | 0.4 |
| Mineral mix | 0.3 | 0.3 | 0.3 | 0.3 |
| Taurine | 1 | 1.1 | 1.1 | 1.1 |
| Stabilized vitamin C | 0.05 | 0.05 | 0.05 | 0.05 |
| Choline chloride | 0.05 | 0.05 | 0.05 | 0.05 |
| Calcium dihydrogen phosphate | 1 | 1 | 1 | 1 |
| Carboxymethyl cellulose | 1 | 1 | 1 | 1 |
| Proximate component (%) | | | | |
| Feeding trial 1 | | | | |
| Moisture | 2.22 | 3.42 | 3.62 | |
| Crude protein | 48.90 | 49.54 | 51.85 | |
| Crude lipid | 17.75 | 18.73 | 16.79 | |
| Ash | 8.78 | 5.85 | 4.90 | |
| NFE + crude fiber * | 22.35 | 22.45 | 22.84 | |
| Feeding trial 2 | | | | |
| Moisture | 7.57 | 6.56 | 8.31 | 8.83 |
| Crude protein | 46.58 | 46.14 | 50.26 | 50.32 |
| Crude lipid | 12.55 | 16.14 | 9.10 | 12.26 |
| Ash | 7.92 | 5.91 | 4.93 | 4.93 |
| NFE + crude fiber * | 25.38 | 25.25 | 27.41 | 23.66 |

* NFE + crude fiber = 10 − (moisture + crude protein + crude lipid + ash); Abbreviations: DHA, docosahexaenoic acid; EPA, eicosapentaenoic acid.

### 4.3. Feeding Trials

Animal experiments were carried out in accordance with the guidelines at Ehime University. The study protocol was approved by the Institutional Animal Care and Use Committee (IACUC) of Ehime University (Permit Number: 3908). All body measurements and surgical manipulations were performed under anesthesia with 0.5–1 mL/L 2-phenoxyethanol, and all efforts were made to minimize suffering. Juvenile red seabreams were kindly provided by Yamasaki Giken Co., Ltd. (Susaki, Kochi, Japan), and had been maintained in 1000 L tanks containing natural sea water on a flow-through system with sand filtration. Fish under one year of age with no traumatic injuries or malformations were included in the trials. Unique identification tags were intraperitoneally injected into the fish bodies, and the animals were allocated into different feeding study groups of 22–40 fish. Two tanks per study group containing half of the fish were cultivated in 100 or 200 L round tanks, depending on availability at the time of the trial. In the trials, fish were fed once per day except for Sunday with the test diets by satiation. The water temperature was 14–18 °C in the 70% FM/HF trial, 15–20 °C in 40% FM/HF Trial 1, and 27–29 °C in 40% FM/HF Trial 2. The main trial with the 70% FM/HF diets was initiated after preliminary feeding with subject diets for two weeks. In the feeding trial for the evaluation of catechol, juvenile red seabreams provided by Bio Ehime Co. Ltd. (Imabari, Ehime, Japan) were used. Thirty fish were cultured in a 200 L round tank, and the water temperature was 17–20 °C. All trials were conducted over a one month period. FL and BW were measured every two weeks.

BW gain, BW gain rate, total feed intake per fish, and feed conversion ratio were calculated as follows:

FL gain (mm) = FL at trial end—initial FL,
BW gain (g) = BW at trial end—initial BW,
FL gain percentage (%) = FL gain/initial FL × 100,
BW gain percentage (%) = BW gain/initial BW × 100,
Feed intake per fish (g) = total feed intake per group (g)/number of fish,
Feed conversion ratio = total feed intake per group (g)/BW gain per group.

Measurements in individual fish were used to obtain the FL, BW, FL gain, BW gain, FL gain rate, and BW gain rate in the feeding test groups, and values in the duplicate tanks were used to obtain the total feed intake per fish and feed conversion ratio in each study group.

### 4.4. Histological Observations

At the end of the feeding trial for the evaluation of catechol, intestines excised from red seabream were fixed in Davidson's solution and embedded in paraffin wax. Sections were cut at 5 μm thickness and stained with hematoxylin and eosin. Morphological differences were evaluated with measurement of microvillus heights, epithelial-cell heights, and goblet-cell areas using a phase-contrast microscope (BX51N-33-PH, Olympus, Tokyo, Japan), and ImageJ (National Institutes of Health, Bethesda, MD, USA). Five heights or areas in each intestine from five fish per group were measured (n = 25/group).

### 4.5. Analysis of Free Catechol in Housefly Larvae

To analyze the free catechol in HF larvae, 0.5 g of samples was mixed well in 1 mL water. Subsequently, 5 mL of acetone was added, and samples were incubated for 30 min at room temperature. The collected supernatants after centrifugation were analyzed by GC/MS analysis using the Agilent 7890 gas chromatography system (Agilent Technologies, Santa Clara, CA, USA) equipped with a CP-Sil8-CB capillary column (30 m × 0.25 mm × 0.25 μm; Agilent Technologies, Santa Clara, CA, USA) connected to a JMS-Q1500 mass spectrometer (JEOL, Tokyo, Japan). Helium was used as the carrier gas at a constant flow of 1.2 mL/min. Column temperature conditions were as follows: 100 °C (1 min) −10 °C/min − 200 °C (10 min) – 40 °C/min −300 °C (10 min). MS parameters for m/z 63, 64, 81, 92, and

110 were obtained using SIM mode and the electron-ionization method. Pyrocatechol (Sigma-Aldrich, St. Louis, MO, USA) was used as the standard.

### 4.6. Proximate-Composition, Amino Acid, and Fatty Acid Analysis

The proximate composition, amino acids, and fatty acids were analyzed using the Association of Analytical Communities (AOAC) methods [47]. The content of crude protein was analyzed using the Kjeldahl method. "Kjeltab" (containing $K_2SO_4$) was added to the ground samples, and samples were digested in a block heater (Tecator TM Digestion Systems 2520, FOSS, Hillerød, Denmark). Nitrogen content was automatically analyzed using an autoanalyzer (Kjeltec$^{TM}$ 8400, FOSS, Hillerød, Denmark). The nitrogen–protein conversion factor was 6.25 for the calculation of crude protein from the nitrogen content. Crude fat was analyzed using the Soxhlet extraction method. Petroleum ether was used as the solvent for extraction from the ground samples in an automated extractor (Soxtec$^{TM}$ 8000, FOSS, Hillerød, Denmark). Ash was analyzed using an electric furnace (MMF-1, AS ONE, Osaka, Japan).

Proteinogenic amino acids in the meal samples were analyzed using an automated amino acid analyzer (Shimadzu, Kyoto, Japan) after hydrochloric hydrolysis with sodium chloride. For methionine and cystine, samples were oxidized with performic acid prior to hydrochloric hydrolysis. For tryptophan, samples were prepared with barium hydroxide octahydrate and thiodietylene glycol before hydrolysis with sodium chloride and analyzed with high-performance liquid chromatography. Taurine was measured using high-performance liquid chromatography. Fatty acids in samples were prepared with hydrolysis and extracted with diethyl ether-petroleum ether (1:1). Gas chromatograph GC-2010 (Shimadzu Corporation, Kyoto, Japan) and Agilent J&W column DB-23 (Agilent Technologies Inc., Santa Clara, CA, USA) were used for analysis with a method described in food-labelling standards by the Consumer Affairs Agency of Japan (CAA, 2015). Fatty acids were identified with Supelco 37 Component FAME Mix (Sigma-Aldrich Co. LLC, St. Louis, MO, USA). Analysis of proteinogenic amino acids, taurine, and fatty acids was conducted by the Japan Food Research Laboratories (Osaka, Japan).

### 4.7. Statistical Analysis

The statistical methods and *p*-values in the present study are shown in the footnotes of the figures and tables. All tests were conducted with the "R" software (https://www.r-project.org).

## 5. Conclusions

The removal of hydrophobic fractions from housefly larvae enabled the complete replacement of FM in the diet of red seabream in the feeding trials. However, HF larvae that were supplemented with the hydrophobic fractions resulted in a significant growth reduction despite the dietary n-3 HUFA levels meeting their nutrition requirements. Since dietary intake of catechol detected in the hydrophobic fraction from HF larvae showed growth reduction and morphological alteration in the intestine, our findings indicate that hydrophobic fractions from insect larvae contain a negative factor for fish growth and eliminating the fraction from the insects is thought to be an important process.

**Author Contributions:** A.I., T.O., C.M., and T.M. planned the projects and conceived the experiments. A.H., A.I., T.O., S.T.T., and R.M. performed the animal experiments. A.H., A.I., T.O., and T.M. analyzed the amino acids, fatty acids, and proximate compositions in the test diets. A.H. and T.O. analyzed the HF larvae by GC/MS. C.M., M.N., and T.M. conducted the histological analysis. A.I. coordinated the supply of the HF larvae. T.T. designed the basic composition of the experimental diets. A.H., A.I., and T.M. wrote the manuscript. All authors approved the manuscript. A.H., A.I., and T.O. made equal contributions to this study.

**Funding:** This research was funded by the A-Step Promoting R and D program from the Japan Science and Technology Agency, a grant-in-aid for Scientific Research from the Japan Society for the Promotion of Science (Number: 26310310), and by the Shintoa Corporation.

**Acknowledgments:** We thank the Yamasaki Giken Co., Ltd. for providing the experimental fish, E's Inc. for providing the housefly larvae, the Shintoa Corporation for providing the meal-worm larvae, the Bizen Chemical Co., Ltd. for providing DHA and EPA, T. Iwai for the technical advice, and K. Kozuki, K. Komatsu, M. Kuwahara,

R. Wada, and Y. Kajiwara for their technical assistance. This study was supported by the Japan Science and Technology Agency, and by funding from the Shintoa Corporation.

**Conflicts of Interest:** The authors declare no conflict of interest. The funders had no role in the design of the study, in the collection, analyses, or interpretation of data, in the writing of the manuscript, or in the decision to publish the results.

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
