# Peer review of "Housefly (Musca domestica) Larvae Preparations after Removing the Hydrophobic Fraction Are Effective Alternatives to Fish Meal in Aquaculture Feed for Red Seabream (Pagrus major)"

_fishes, doi:10.3390/fishes4030038_

Round 1
Reviewer 1 Report
The present MS deals with a new approach for the use of M. domestica meal in fish feed preparation by removing the hydrophobic fraction.
The topic is of interest and the MS is generally well presented. however, there is a need for an English revision.
The introduction need a section regarding the possible effects of insect meal inclusion in feeds on the gastro-intestinal tract.
I suggest the authors to read and add these two papers that can assist them in adding a good paragraph about the effects on the intestine by insect meal:
Cutrignelli, Monica Isabella, Messina, Maria, Tulli, Francesca, Randazzi, Basilio, Olivotto, Ike, Gasco, Laura, Loponte, Rosa, Bovera, Fulvia (2018). Evaluation of an insect meal of the Black Soldier Fly ( Hermetia illucens ) as soybean substitute: Intestinal morphometry, enzymatic and microbial activity in laying hens. RESEARCH IN VETERINARY SCIENCE, vol. 117, p. 209-215, ISSN: 0034-5288, doi: 10.1016/j.rvsc.2017.12.020
Giorgini, Elisabetta, Randazzo, Basilio, Gioacchini, Giorgia, Cardinaletti, Gloriana, Vaccari, Lisa, Tibaldi, Emilio, Olivotto, Ike (2018). New insights on the macromolecular building of rainbow trout ( O. mykiss ) intestine: FTIR Imaging and histological correlative study. AQUACULTURE, vol. 497, p. 1-9, ISSN: 0044-8486, doi: 10.1016/j.aquaculture.2018.07.032
Line 35 please add also a reference about ornamental aquaculture including for example this paper:
Vargas-Abúndez, Arturo Jorge, Randazzo, Basilio, FODDAI, MARCO, SANCHINI, LORENZO, Truzzi, Cristina, Giorgini, Elisabetta, Gasco, Laura, Olivotto, Ike (2019). Insect meal based diets for clownfish: Biometric, histological, spectroscopic, biochemical and molecular implications. AQUACULTURE, vol. 498, p. 1-11, ISSN: 0044-8486, doi: 10.1016/j.aquaculture.2018.08.018
L 49 fatty acid requirements should be stated here as well.
Author Response
Reviewer1
We wish to express our deep appreciation for your suggestion. Also, we appreciate your presentation of appropriate articles to improve our manuscript. We highlighted the changes in underline in the revised manuscript.
Reviewer’s comments:
>The topic is of interest and the MS is generally well presented. however, there is a need for an English revision.
Our response:
We improved the manuscript in accordance with your instruction, and the manuscript was edited by MDPI English Editing Service.
Reviewer’s comments:
> The introduction need a section regarding the possible effects of insect meal inclusion in feeds on the gastro-intestinal tract.
Our response:
We added a sentence regarding the dietary effect of insect meal on the intestine of laying hens. Since FT-IR imaging of intestine (Giorgini, et al. 2018) spectroscopy will be needed in further research, we added a sentence regarding the method in L224-225 on Discussion section.
Reviewer’s comments:
>Line 35 please add also a reference about ornamental aquaculture including for example this paper:
Our response:
According to your comments, we added the sentence about a need for substitution for fish meal in ornamental aquaculture in L40-41.
Reviewer’s comments:
> L 49 fatty acid requirements should be stated here as well.
Our response:
We added a sentence regarding the causes including an imbalanced fatty acid profile as stated by the authors (Lin, et al. 2017) in L51-52
Thank you once again for your consideration of our manuscript.
Sincerely yours,

Reviewer 2 Report
Minor comment:
The results showed that the removal of the hydrophobic fractions from housefly larvae enabled complete replacement of FM in the diet for red seabream in the feeding trials and could be a potentially viable option leading to a more environmentally sustainable aquaculture industry in the future. However, a discussion concerning the comparison of costs between FM and HF larvae (including defatting process) is suggested.
Author Response
Reviewer2
Thank you for your useful suggestions.
Reviewer’s comments:
> a discussion concerning the comparison of costs between FM and HF larvae (including defatting process) is suggested.
Our response:
According to your comment, we added a sentence regarding the costs of FM and HF larvae in L173-177 on the Discussion section.
Thank you once again for your consideration of our manuscript.
Sincerely yours,

Reviewer 3 Report
Letter to Authors
fishes-486338
Housefly (Musca domestica)
larvae preparations removal hydrophobic fraction are effective
alternatives to fish meal in aquaculture feed for seabream (Pagrus
major)
Atsushi Hashizume, Atsushi Ido, Takashi Ohta, Serigne Thierno
Thiaw, Ryusaku Munenori Nishikawa, Takayuki Takahashi, Chiemi Miura,
Takeshi Miura
190510
A new trial to reduce fish meal in
aquaculture diets replacing by maggot is interesting and worth
publishing in the journal you are submitting to. This MS contains,
however, a major issue regarding role of catechol in fish growth
reduction. You emphasize that catechol reduces the bream growth
performance, but this discussion has no rationale. Experimental
conditions employed here to show growth deficiency was unpractical
(>=200ug/g). You actually say other hydrophobic molecules seems to
effect fish growth. You should omit statements regarding catechol, or
addition of experiments seeking what hydrophobic fraction takes effect.
Hence I am sorry to say I suggested the editor a major revision process.
Other minor issues also exist. Method description is awkward.
Terminology of HF, HF larvae, HF meal, and HF diet is particularly
confusing. See below for detail.
L44
No study conducted on insectivorous fish such as masu salmon parr?
L66
Catechol does not affect growth reduction at a practical condition (~20ug/g). See L192.
L77 Table 1
Ise?
L94
with or without the hydrophobic fraction (HF-HPF and HF+HPF) ->with or without the hydrophobic fraction (HF+HPF and HF+-HPF)?
L115,118,122,151
Caption of tables should be put over the body.
L131
was less -> declined
L135
it can be seen that (redundant and weak) -> delete
L155
significantly -> successfully or delete
L169
Fish has gastric chitinase (e.g. Ikeda et al 2017).
L186-201
Irrational or inconsistent statements.
L202
in the absence of -> without
L221
To defat HF larvae?
This
implicitly tells that you prepared specifically HF-HPF by defatting,
and HF+HPF was not undergone the defatting process, but see L239.
Did you prepared HF+HPF and HF-HPF larvae with or without adding HPF fraction to defatted HF larvae? If so;
To defat HF larvae -> delete
ground HF larvae were treated -> Ground HF larvae were then treated
L236
study -> {experiment, examination, trial, attempt}
L238
7% fish oil was added -> Fish oil was added at 7%
Do not begin with numbers.
L241
more 2 -> two more
Spell out numbers equal to or lower than ten.
L243
and undefatted HF+DHA+EPA?
You
already added DHA and EPA (HF+DHA+EPA). Then n-3 highly unsaturated
fatty acid levels of undefatted HF supplemented with DHA and EPA are
always equal to HF+DHA+EPA at any dose.
L273
under anesthesia with 0.5 - 1 mL/L 2-phenoxyethanol (redundant) -> delete
L274
Do not begin with numbers.
Spell out bear numbers equal to or lower than ten.
L282
under anesthesia (redundant) -> delete
L363 references.
Check
the list carefully from the beginning. It is the authors'
responsibility that all references are properly cited. You might
add/remove/swap references during internal reviewing. Numbering of the
list should then be particularly checked to avoid irrational citation.
The following literature may be helpful for your further discussion.
Ikeda M, Kakizaki H, Matsumiya M. 2017. Biochemistry of fish stomach chitinase. Int J Biol Macromol 104B:1672-1681.
Author Response
Reviewer3
Thank you for providing useful comments. We improved the manuscript in accordance with your instruction, and the manuscript was edited by MDPI English Editing Service. We highlighted the changes in underline in the revised manuscript.
Reviewer’s comments:
>Experimental conditions employed here to show growth deficiency was unpractical (>=200ug/g). You actually say other hydrophobic molecules seems to effect fish growth. You should omit statements regarding catechol, or addition of experiments seeking what hydrophobic fraction takes effect.
Our response:
As you mentioned, catechol is not only the causative substance of the growth reduction in the hydrophobic fraction from HF larvae, but other molecules than catechol in the fraction seem to affect negatively on fish growth. However, we found that same level of dietary catechol in feed 70% HF larvae (20 μg/g) give an alteration in the intestine of red seabream. Since we believe that dietary effect of catechol in fish is an important finding in the field of research about insect for feed, we cannot omit the statement of catechol, but we have rewritten L210-215 in the Discussion section with careful consideration of your comment. We hope that you agree.
Reviewer’s comments:
> L44: No study conducted on insectivorous fish such as masu salmon parr?
Our response:
As far as we know, there is no study about dietary housefly with salmonids.
Reviewer’s comments:
> L66: Catechol does not affect growth reduction at a practical condition (~20ug/g). See L192.
Our response:
To clarify it, we added the sentence in L72-74, “Although no significant difference in growth was observed with the dietary low level of catechol, morphology alterations were observed in the intestine of red seabream.”
Reviewer’s comments:
> L77 Table 1: Ise?
Our response:
We corrected our mistake. We changed “Ise” into “Ile”.
Reviewer’s comments:
> L94:with or without the hydrophobic fraction (HF-HPF and HF+HPF) ->with or without the hydrophobic fraction (HF+HPF and HF+-HPF)?
Our response:
We corrected it according to your comment.
Reviewer’s comments:
> L115,118,122,151: Caption of tables should be put over the body.
Our response:
We corrected it according to your comment.
Reviewer’s comments:
> L131: was less -> declined
Our response:
We changed it as your comment.
Reviewer’s comments:
> L135: it can be seen that (redundant and weak) -> delete
Our response:
We deleted it.
Reviewer’s comments:
> L155: significantly -> successfully or delete
Our response:
We changed it into “successfully”.
Reviewer’s comments:
> L169: Fish has gastric chitinase (e.g. Ikeda et al 2017).
Our response:
Thanks for giving us an appropriate article to improve our discussion. We added the sentence about chitinase in the stomach with the citation in L193.
Reviewer’s comments:
> L186-201: Irrational or inconsistent statements.
Our response:
As we mentioned above, we believe that we should not omit the statement of catechol, but we have rewritten so in Discussion L214-220. We hope that you agree.
Reviewer’s comments:
>L202: in the absence of -> without
Our response:
We corrected as your instruction.
Reviewer’s comments:
>L221: To defat HF larvae?
>This implicitly tells that you prepared specifically HF-HPF by defatting, and
>HF+HPF was not undergone the defatting process, but see L239.
>Did you prepared HF+HPF and HF-HPF larvae with or without adding HPF
>fraction to defatted HF larvae? If so;
>To defat HF larvae -> delete
>ground HF larvae were treated -> Ground HF larvae were then treated
Our response:
We apologize to bring your confusion.
HF+HPF in L239 (L265 in the revised manuscript) is a formulated diet including defatted HF larvae and HPF obtained in the defatting process. HF-HPF in L264 in the revised manuscript is also diet including defatted HF larvae and fish oil. We should not have used the name “HF-HPF” with defatted HF larvae which are feed ingredients, not diets. Therefore, we changed “HF-HPF” into “defatted HF larvae” in Table 7.
Reviewer’s comments:
>L236: study -> {experiment, examination, trial, attempt}
Our response:
We changed it into “trial”.
Reviewer’s comments:
>L238: 7% fish oil was added -> Fish oil was added at 7% Do not begin with numbers.
Our response:
We corrected it as your instruction.
Reviewer’s comments:
>L241: more 2 -> two more. Spell out numbers equal to or lower than ten.
Our response:
We corrected it as your instruction.
Reviewer’s comments:
>L243: and undefatted HF+DHA+EPA?
>You already added DHA and EPA (HF+DHA+EPA). Then n-3 highly unsaturated fatty acid levels of undefatted HF supplemented with DHA and EPA are always equal to HF+DHA+EPA at any dose.
Our response:
We apologize to bring your confusion again.
HF+DHA+EPA is not a feed ingredient, but a formulated diets shown in table 9. We didn’t add DHA+EPA to HF larvae, but we added them to the diets as shown in L268.
Reviewer’s comments:
> L273: under anesthesia with 0.5 - 1 mL/L 2-phenoxyethanol (redundant) -> delete
Our response:
Another reviewer told us to specify the amount of anesthetic. We hope you can agree.
Reviewer’s comments:
>L274: Do not begin with numbers. Spell out bear numbers equal to or lower than ten.
Our response:
We corrected it as your instruction.
Reviewer’s comments:
> L282: under anesthesia (redundant) -> delete
Our response:
We deleted it.
Reviewer’s comments:
> L363 references. Check the list carefully from the beginning. It is the authors' responsibility that all references are properly cited. You might add/remove/swap references during internal reviewing. Numbering of the list should then be particularly
checked to avoid irrational citation.
Our response:
We carefully checked the reference list.
Thank you once again for your consideration of our manuscript.
Sincerely yours,

Reviewer 4 Report
The study is interesting and experimental design properly planned. Laboratory analysis carried out cover different aspects (nutritional, histological, etc.) on the basis of which discussion is well argued.
On the other hand, the paper at present needs a thorough editing for English language usage.
Specific comments:
1) In M&M section fatty acids profile analysis is not reported. What method was used to extract lipid fraction? The same used for crude fat quantification (Soxhlet?) or Folch? And for the lipids characterization? Gas chromatographic analysis? what kind of column was used?
2) Fatty acid analysis on HF larvae showed a very high fraction of "not identified FAs" (12.4%). Why? this percentage should not be accepted or it requires an explanation.
Author Response
Reviewer4
Thank you for valuable comments, and giving us an opportunity to improve the manuscript. We highlighted the changes in underline in the revised manuscript.
Reviewer’s comments:
>On the other hand, the paper at present needs a thorough editing for English language usage.
Our response:
We improved the manuscript in accordance with your instruction, and the manuscript was edited by MDPI English Editing Service.
Reviewer’s comments:
> 1) In M&M section fatty acids profile analysis is not reported. What method was used to extract lipid fraction? The same used for crude fat quantification (Soxhlet?) or Folch? And for the lipids characterization? Gas chromatographic analysis? what kind of column was used?
Our response:
As your instruction, we rewrote the section in L357-361, as “Fatty acids in samples were prepared with hydrolysis and extracted with diethyl ether - petroleum ether (1:1). Gas chromatograph GC-2010 (Shimadzu Corporation, Kyoto, Japan) and Agilent J&W column DB-23 (Agilent Technologies, Inc., Santa Clara, CA) were used for analysis with a method described in food-labelling standards by the Consumer Affairs Agency of Japan (CAA, 2015). Fatty acids were identified with Supelco 37 Component FAME Mix (Sigma-Aldrich Co. LLC, St. Louis, MO).”
Reviewer’s comments:
> 2) Fatty acid analysis on HF larvae showed a very high fraction of "not identified FAs" (12.4%). Why? this percentage should not be accepted or it requires an explanation.
Our response:
The HF larvae contained 12.4% FAs which we cannot detect with the above mentioned method. Not only this study, there are other studies showing high levels of unidentified FA in HF larvae. We added the sentence regarding the unidentified FAs in HF larvae in L183-188, as “12.4% of total fatty acids were not identified in this study. Unidentified fatty acids in HF larvae or pupae were relatively high (7.2-8.5%) comparing with other insects [19]. Since Fitches (2018) showed that composition of unidentified fatty acids in HF larvae reached 15% depended on sources of poultry litter as rearing substrates for HF larvae [12], these unidentified fatty acids seem to be derived from pig manure used for rearing the larvae.”
Thank you once again for your consideration of our manuscript.
Sincerely yours,

Reviewer 5 Report
This study investigated the effects of dietary housefly larvae meal with or without removal hydrophobic fraction on the growth performance of red seabream, Pagrus major. This study had two experiments. The first trial, the treatment diets containing 70% fish meal with 7% fish oil, 70% defatted housefly larvae meal (HF) with 7% fish meal and 70% original HF meal without adding fish oil. The second trial, the treatment diets containing 40% fish meal with 11% fish oil, 40% defatted HF with 12% fish oil, 40% original HF and 40% original HF with EPA and DHA.
1. Line 241 Authors produced two more diets ----.
2. Line 223-254 Authors issued that “Catechol was added to the diets to be --- and the surface of the diets was coated with plant-derived oil”. It is hard to understand this feeding trial designation. Authors need to provide what the basal diet is for this trial.
3. The term of BW gain ratio is wrong. It should be corrected as weight gain percentage.
4. Three feeding trails were conducted in different water temperatures. They cannot compare the growth performance each other.
5. Since authors analyzed concentration of catechol in this study, it is better for authors to provide the concentration of catechol of treatment diets.
6. Authors need to explain why experiments 1 and 2 used different levels of FM or HF.
7. The treatment diets in experiment 1 and 2 showed different protein levels ranging from 46-53.9%. They are not isonitrogenous diets.
8. Fig. 1 can be deleted.
9. Line 91-106 Authors can provide more details in the result part concerning the growth performance of red seabream.
Author Response
Reviewer5
Thank you for providing the useful comments. We improved the manuscript in accordance with your instruction, and the manuscript was edited by MDPI English Editing Service. We highlighted the changes in underline in the revised manuscript.
Reviewer’s comments:
> 1. Line 241 Authors produced two more diets ----.
Our response:
We corrected it according to your instruction.
Reviewer’s comments:
> 2. Line 223-254 Authors issued that “Catechol was added to the diets to be --- and the surface of the diets was coated with plant-derived oil”. It is hard to understand this feeding trial designation. Authors need to provide what the basal diet is for this trial.
Our response:
We showed that the basal diets named “Otohime” is commercially available in L277-278.
Reviewer’s comments:
> 3. The term of BW gain ratio is wrong. It should be corrected as weight gain percentage.
Our response:
We changed “gain ratio” into “gain percentage”.
Reviewer’s comments:
> 4. Three feeding trails were conducted in different water temperatures.
They cannot compare the growth performance each other.
Our response:
We set control group in every feeding trials. It is possible to compare the test groups and control group in each trials even in different water temperatures.
Reviewer’s comments:
> 5. Since authors analyzed concentration of catechol in this study, it is
better for authors to provide the concentration of catechol of treatment
diets.
Our response:
We agree you comment. Sorry, but we have not analyzed the diet.
Reviewer’s comments:
> 6. Authors need to explain why experiments 1 and 2 used different
levels of FM or HF.
Our response:
Our study showed that various level of FM could be replaced with HF. We think these results are important in the field of research about insects for feed.
Reviewer’s comments:
> 7. The treatment diets in experiment 1 and 2 showed different protein levels ranging from 46-53.9%. They are not isonitrogenous diets.
Our response:
As your comment, we should have design the formulation of these diets to make similar levels of crude protein. However, our findings that HF-HPF can be substitute completely for FM in the control diets must be important in this research field. In table 8, crude protein of control is 51.2%, while that of HF-HPF is 48.9%. In table 9, CP of control is 46.6%, while that of HP-HPF is 46.14%. Therefore, there are not big differences in CP between control and HP-HPF in each feeding trials. Consequently, the difference of CP you mentioned did not affect our conclusion. We hope you agree.
Reviewer’s comments:
> 8. Fig. 1 can be deleted.
Our response:
Figure 1 illustrates the result of the feeding trial, and it makes readers’ understanding easy. We hope you agree that.
Reviewer’s comments:
> 9. Line 91-106 Authors can provide more details in the result part
concerning the growth performance of red seabream.
Our response:
We rewrote the section according to your comments.
Thank you once again for your consideration of our manuscript.
Sincerely yours,

Round 2
Reviewer 3 Report
Letter to Authors
fishes-486338v2
Housefly (Musca domestica) larvae preparations removal hydrophobic fraction are effective alternatives to fish meal in aquaculture feed for seabream (Pagrus major)
Atsushi Hashizume, Atsushi Ido, Takashi Ohta, Serigne Thierno Thiaw, Ryusaku Munenori Nishikawa, Takayuki Takahashi, Chiemi Miura, Takeshi Miura
190510
You have paid an effort to improve your MS, but it is not enough. Your writing style is not very much fine. I am sorry to say a moderate revision process is still necessary.
See below for detail.
L31 keyword
housefly, red seabream, fish meal -> delete
Do not list words which appear also in the title. Duplicate hits upon computer search do not make sense. Give words that do not appear in the title to draw attention from wider readership. Posting words that neither appear in the abstract is better, because indexing robots may not weigh much on words deeper (posterior) in the text even in full-text search platforms. Hint; fatty acid, histology of intestine, substitute diet, sustainable aquaculture, waste recycling, chitin, etc.
L120
a, fork length gain rate; b, body weight gain rate -> Left, fork length gain rate; right, body weight gain rate
L120,121
HF -HPF (en-dash) -> HF-HPF (minus/hyphen)
En-dash indicates a range.
L129,133
Feeding Trial -> lower case
L180
With respect to fatty acids -> new paragraph
You confess a different story begins 'with respect to'.
L181
as a supplement to .. were insufficient. -> as a supplement to the diet. However, recovery of fish growth was partial, and improvements in BW gain and feed intake were insufficient?
Logical linking is unclear.
L182
12.4% -> A {substantial, considerable} portion (12.4%)
Do not begin a sentence with bare numbers.
L184
Fitches (2018) showed that (redundant) -> delete
L187
beta(1 -4) (en-dash) -> beta(1-4) (hyphen/minus)
A hyphen connects words.
L190
Kono (1987) indicated that -> delete
L193
were thought to (irresponsible, who thought that?) -> {may, would, possibly}
L208
growth reduction and a morphological alteration in their intestines (cheat!) -> intestinal morphological alteration in lower dose and growth reduction in higher dose
L221
morphological changes .. provoke infectious diseases (irrational)
The morphological changes could either raise or down disease resistance. You are correct to write morphological alteration or change, not deformity.
L226,227
housefly -> HF
L297
under anesthesia with 0.5-1 mL/L 2-phenoxyethanol -> delete
You have already stated this at L292. If injection of tags is not a surgical manipulation, it is OK though.
Author Response
Thanks for your valuable comments. We revised our manuscript in consideration of your comments. We highlighted the changes in underline in the revised manuscript.
Reviewer’s comments:
>L31 keyword housefly, red seabream, fish meal -> delete
Our response:
We removed housefly, red seabream and fish meal from the keyword list, and added sustainable aquaculture, fish meal replacement, insect for feed.
Reviewer’s comments:
>L120 a, fork length gain rate; b, body weight gain rate -> Left, fork length gain rate; right, body weight gain rate
Our response:
We changed them according to your instruction.
Reviewer’s comments:
>L120, 121 HF -HPF (en-dash) -> HF-HPF (minus/hyphen)
Our response:
We changed “en-dash” into “minus/hyphen”
Reviewer’s comments:
>L129, 133 Feeding Trial -> lower case
Our response:
We changed “Feeding Trial” into “feeding trial”.
Reviewer’s comments:
>L180 With respect to fatty acids -> new paragraph
Our response:
We set a new paragraph.
Reviewer’s comments:
>L181 as a supplement to .. were insufficient. -> as a supplement to the diet. However, recovery of fish growth was partial, and improvements in BW gain and feed intake were insufficient?
Our response:
We corrected it, as “as a supplement to the diet. However, recovery of fish growth was partial, and improvements in BW gain and feed intake were insufficient.”
Reviewer’s comments:
>L182 12.4% -> A {substantial, considerable} portion (12.4%)
Our response:
We rewrote as your instruction as “A substantial portion (12.4%) of…”
Reviewer’s comments:
>L184 Fitches (2018) showed that (redundant) -> delete
Our response:
We deleted it as your instruction.
Reviewer’s comments:
>L187 beta(1 -4) (en-dash) -> beta(1-4) (hyphen/minus)
Our response:
We corrected it as your instruction.
Reviewer’s comments:
>L190 Kono (1987) indicated that -> delete
Our response:
We deleted it as your instruction.
Reviewer’s comments:
>L193 were thought to (irresponsible, who thought that?) -> {may, would, possibly}
Our response:
We changed “were thought to” into “may” as your instruction.
Reviewer’s comments:
>L208 growth reduction and a morphological alteration in their intestines (cheat!) -> intestinal morphological alteration in lower dose and growth reduction in higher dose
Our response:
We corrected it as your instruction.
Reviewer’s comments:
>L221 morphological changes .. provoke infectious diseases (irrational)
Our response:
We changed “changes” into “alterations” according to your comment.
Reviewer’s comments:
>L226, 227 housefly -> HF
Our response:
We corrected them as your instruction
Reviewer’s comments:
>L297 under anesthesia with 0.5-1 mL/L 2-phenoxyethanol -> delete
Our response:
We deleted it.
Thank you once again for your consideration of our manuscript.
Sincerely yours,
